# Adaptive Algorithms for Relaxed Pareto Set Identification

**Cyrille Kone**[1]
cyrille.kone@inria.fr

**Emilie Kaufmann**[1]
emilie.kaufmann@univ-lille.fr

**Laura Richert**[2]
laura.richert@u-bordeaux.fr

[1] Univ. Lille, Inria, CNRS, Centrale Lille, UMR 9198-CRIStAL, F-59000 Lille, France
[2] Univ. Bordeaux, Inserm, Inria, BPH, U1219, Sistm, F-33000 Bordeaux, France

## Abstract

In this paper we revisit the fixed-confidence identification of the Pareto optimal set in a multi-objective multi-armed bandit model. As the sample complexity to identify the exact Pareto set can be very large, a relaxation allowing to output some additional near-optimal arms has been studied. In this work we also tackle alternative relaxations that allow instead to identify a relevant *subset* of the Pareto set. Notably, we propose a single sampling strategy, called Adaptive Pareto Exploration, that can be used in conjunction with different stopping rules to take into account different relaxations of the Pareto Set Identification problem. We analyze the sample complexity of these different combinations, quantifying in particular the reduction in sample complexity that occurs when one seeks to identify at most $k$ Pareto optimal arms. We showcase the good practical performance of Adaptive Pareto Exploration on a real-world scenario, in which we adaptively explore several vaccination strategies against Covid-19 in order to find the optimal ones when multiple immunogenicity criteria are taken into account.

## 1 Introduction

In a multi-armed bandit model, an agent sequentially collects samples from several unknown distributions, called arms, in order to learn about these distributions (pure exploration), possibly under the constraint to maximize the samples collected, viewed as rewards (regret minimization). These objectives have been extensively studied for different types of univariate arms distributions [21]. In this paper, we consider the less common setting in which arms are multi-variate distributions. We are interested in the *Pareto Set Identification* (PSI) problem. In this pure exploration problem, the agent seeks to identify the arms that are *(Pareto) optimal*, i.e. such that their expected values for all objectives are not uniformly worse than those of another arm.

We formalize this as a fixed-confidence identification problem: in each round $t$ the agent selects an arm $A_t$ using an adaptive *sampling rule* and observes a sample $\mathbf{X}_t \in \mathbb{R}^D$ from the associated distribution. It further uses an adaptive stopping rule $\tau$ to decide when to stop sampling and output a set of arms $\widehat{S}_\tau$ which is her guess for (an approximation of) the true Pareto set $\mathcal{S}^\star$. Given a risk parameter $\delta \in (0, 1)$, this guess should be correct with high probability, e.g. satisfy $\mathbb{P}(\widehat{S}_\tau = \mathcal{S}^\star) \geq 1 - \delta$ for exact Pareto set identification, while requiring a small *sample complexity* $\tau$. This generalizes the well-studied fixed-confidence Best Arm Identification (BAI) problem [8, 15, 10] to multiple objectives.

37th Conference on Neural Information Processing Systems (NeurIPS 2023).

Our motivation to study multi-objective adaptive identification stems from the design of adaptive early-phase clinical trials. In phase I/II trials, the effects of a molecule in humans are explored, and several biological endpoints may be assessed at the same time as indicative markers of efficacy. In particular, in the context of vaccine development, early-phase trials usually assess multiple immunogenicity endpoints (i.e. various markers of the effects of the vaccine on the immune system, such as different facets of antibody responses or other immune parameters). In the absence of a known correlate of protection during early clinical development, these endpoints may not have a clear *a priori* hierarchy, may not all be correlated, which makes an examination of the Pareto set of different vaccinal strategies particularly relevant. In addition, given the availability of various vaccine platforms (such as mRNA vaccines, viral-vector vaccines, protein vaccines), as exemplified by Covid-19 vaccines, there may be a need to adaptively screen the various resulting vaccine strategies to select the most promising ones. Apart from clinical trials, active Pareto Set Identification can be meaningful in many real-word contexts, and we refer the reader to the various examples given by [28], such as in hardware or software design. Other applications include A/B/n testing for marketing or online recommender systems in which it is common to jointly optimize multiple (possibly conflicting) objectives such as user behavioral metrics (e.g. clicks, streams, dwell time, etc), supplier exposure objectives (e.g. diversity) and platform centric objectives (e.g. promotions) [22].

For many applications, the sample complexity of exact PSI can be prohibitive, either when there are many close to optimal arms or when the Pareto set is very large, and different relaxations have been considered in the literature [4, 27]. Going back to our motivation, in an adaptive trial that aims at pre-selecting a certain number of treatments or vaccine strategies for further investigations in clinical trials, practical constraints (the cost and feasibility of the trials) impose a constraint on the maximal number of interesting arms that can be identified. This motivates the introduction of a new setting where the agent is asked to identify *at most* $k$ Pareto optimal arms. Interestingly the sampling rule that we propose for this setting can be used to solve (some generalizations of) other relaxations considered in the literature.

**Related work**  The work most closely related to ours is that of Auer et al. [4], who propose a relaxation, which we refer to as $\varepsilon_1$-PSI: their algorithm returns a set $\widehat{S}$ that contains w.h.p. all the Pareto optimal arms and possibly some sub-optimal arms, which when increased by $\varepsilon_1$ coordinate-wise become Pareto optimal. For arms that have sub-Gaussian marginals, they provide an instance-dependent sample complexity bound scaling with some notion of sub-optimality gap for each arm. The work of Zaluaga et al. [28, 27] studies a structured variant of fixed-confidence PSI in which the means are regular functions of arms' descriptors. They use Gaussian process modeling and obtain worse-case sample complexity bounds. In particular [27] considers the identification of an $\varepsilon$-cover of the Pareto set, which is a representative subset of the ($\varepsilon$)-Pareto set that will be related to our $(\varepsilon_1, \varepsilon_2)$-PSI criterion. The algorithms of [4] and those of [28, 27] in the unstructured setting[1] have the same flavor: they sample uniformly from a set of active arms and remove arms that have been found sub-optimal (or not representative). Auer et al.[4] further adds an acceptation mechanism to stop sampling some of the arms that have been found (nearly-)optimal and are guaranteed not to dominate an arm of the active set. In this paper, we propose instead a more adaptive exploration strategy, which departs from such accept/reject mechanisms and is suited for different types of relaxation, including our novel $k$-relaxation.

Adaptive Pareto Exploration (APE) leverages confidence intervals on the differences of arms' coordinates in order to identify a single arm to explore, in the spirit of the LUCB [15] or UGapEc [9] algorithms for Top-$m$ identification in (one-dimensional) bandit models. These algorithms have been found out to be preferable in practice to their competitors based on uniform sampling and eliminations [19], an observation that will carry over to APE. Besides the multi-dimensional observations, we emphasize that a major challenge of the PSI problem with respect to e.g. Top $m$ identification is that the number of arms to identify is not known in advance. Moreover, when relaxations are considered, there are multiple correct answers. In the one-dimensional settings, finding optimal algorithms in the presence of multiple correct answers is notoriously hard as discussed by the authors of [5], and their lower-bound based approach becomes impractical in our multi-dimensional setting. Finally, we remark that the $k$-relaxation can be viewed as an extension of the problem of identifying any $k$-sized subset out of the best $m$ arms in a standard bandit [25].

---

[1]PAL relies on confidence intervals that follow from Gaussian process regression, but can also be instantiated with simpler un-structured confidence intervals as those used in our work and in Auer's

Beyond Pareto set identification, other interesting multi-objective bandit identification problems have been studied in the literature. For example [6] propose an algorithm to identify some particular arms in the Pareto set through a scalarization technique [23]. The idea is to turn the multi-objective pure-exploration problem into a single-objective one (unique optimal arm) by using a real-valued preference function which is only maximized by Pareto optimal arms (see e.g [23] for some examples of these functions). In practice, a family of those functions can be used to identify many arms of the Pareto set but it is not always possible to identify the entire Pareto set using this technique (see e.g [7] for *weighted sum* with a family of weights vectors). In a different direction, the authors of [16] introduce the feasible arm identification problem, in which the goal is to identify the set of arms whose mean vectors belong to a known polyhedron $P \subset \mathbb{R}^D$. In a follow up work [17], they propose a fixed-confidence algorithm for finding feasible arms that further maximize a given weighted sum of the objectives. In clinical trials, this could be used to find treatments maximizing efficacy (or a weighted sum of different efficacy indicators), under the constraint that the toxicity remains below a threshold. However, in the presence of multiple indicators of biological efficacy, choosing the weights may be difficult, and an examination of the Pareto set could be more suitable. Finally, some papers consider extensions of the Pareto optimality condition. The authors of [1] tackle the identification of the set of non-dominated arms of any partial order defined by an $\mathbb{R}^D$ polyhedral ordering cone (the usual Pareto dominance corresponds to using the cone defined by the positive orthant $\mathbb{R}^D_+$), and they provide worst-case sample complexity in the PAC setting. The work of [3] studies the identification of the set of non-dominated elements in a *partially ordered set* under the dueling bandit setting, in which the observations consists in pairwise comparison between arms.

**Outline and contributions**    First, we formalize in Section 2 different relaxations of the PSI problem: $\varepsilon_1$-PSI, as introduced by [4], $\varepsilon_1, \varepsilon_2$-PSI, of which a particular case was studied by [27] and $\varepsilon_1$-PSI-$k$, a novel relaxation that takes as input an upper bound $k$ on the maximum number of $\varepsilon_1$-optimal arms that can be returned. Then, we introduce in Section 3 Adaptive Pareto Exploration, a simple, adaptive sampling rule which can simultaneously tackle all three relaxations, when coupled with an appropriate stopping rule that we define for each of them. In Section 4, we prove high-probability upper bounds on the sample complexity of APE under different stopping rules. For $\varepsilon_1$-PSI, our bound slightly improves upon the state-of-the-art. Our strongest result is the bound for $\varepsilon_1$-PSI-$k$, which leads to a new notion of sub-optimality gap, quantifying the reduction in sample complexity that is obtained. Then, Section 5 presents the result of a numerical study on synthetic datasets, one of them being inspired by a Covid-19 vaccine clinical trial. It showcases the good empirical performance of APE compared to existing algorithms, and illustrates the impact of the different relaxations.

## 2   Problem Setting

In this section, we introduce the *Pareto Set Identification* (PSI) problem and its relaxations. Fix $K, D \in \mathbb{N}^\star$. Let $\nu_1, \ldots, \nu_K$ be distributions over $\mathbb{R}^D$ with means $\boldsymbol{\mu}_1, \ldots, \boldsymbol{\mu}_K \in \mathbb{R}^D$. Let $\mathbb{A} := [K] := \{1, \ldots, K\}$ denote the set of arms. Let $\nu := (\nu_1, \ldots, \nu_K)$ and $\mathcal{X} := (\boldsymbol{\mu}_1, \ldots, \boldsymbol{\mu}_K)$. We use boldfaced symbols for $\mathbb{R}^D$ elements. Letting $\mathbf{X} \in \mathbb{R}^D, u \in \mathbb{R}$, for any $d \in \{1, \ldots, D\}$, $\mathrm{X}^d$ denotes the $d$-th coordinate of $\mathbf{X}$ and $\mathbf{X} + u := (X^1 + u, \ldots, X^D + u)$. In the sequel, we will assume that $\nu_1, \ldots, \nu_K$ have 1-subgaussian marginals [2].

**Definition 1.** *Given two arms $i, j \in \mathbb{A}$, $i$ is weakly (Pareto) dominated by $j$ (denoted by $\boldsymbol{\mu}_i \leq \boldsymbol{\mu}_j$) if for any $d \in \{1, \ldots, D\}$, $\mu_i^d \leq \mu_j^d$. The arm $i$ is (Pareto) dominated by $j$ ($\boldsymbol{\mu}_i \preceq \boldsymbol{\mu}_j$ or $i \preceq j$) if $i$ is weakly dominated by $j$ and there exists $d \in \{1, \ldots, D\}$ such that $\mu_i^d < \mu_j^d$. The arm $i$ is strictly (Pareto) dominated by $j$ ($\boldsymbol{\mu}_i \prec \boldsymbol{\mu}_j$ or $i \prec j$) if for any $d \in \{1, \ldots, D\}$, $\mu_i^d < \mu_j^d$.*

For $\boldsymbol{\varepsilon} \in \mathbb{R}^D_+$, the $\boldsymbol{\varepsilon}$-*Pareto set* $\mathcal{S}^\star_{\boldsymbol{\varepsilon}}(\mathcal{X})$ is the set of $\boldsymbol{\varepsilon}$-Pareto optimal arms, that is:
$$\mathcal{S}^\star_{\boldsymbol{\varepsilon}}(\mathcal{X}) := \{i \in \mathbb{A} \text{ s.t } \nexists j \in \mathbb{A} : \boldsymbol{\mu}_i + \boldsymbol{\varepsilon} \prec \boldsymbol{\mu}_j\}.$$

In particular, $\mathcal{S}^\star_0(\mathcal{X})$ is called the *Pareto set* and we will simply write $\mathcal{S}^\star(\mathcal{X})$ to denote $\mathcal{S}^\star_0(\mathcal{X})$. When it is clear from the context, we write $\mathcal{S}^\star$ (or $\mathcal{S}^\star_{\boldsymbol{\varepsilon}}$) to denote $\mathcal{S}^\star(\mathcal{X})$ (or $\mathcal{S}^\star_{\boldsymbol{\varepsilon}}(\mathcal{X})$). By abuse of notation we write $\mathcal{S}^\star_{\varepsilon}$ when $\varepsilon \in \mathbb{R}^+$ to denote $\mathcal{S}^\star_{\boldsymbol{\varepsilon}}$, with $\boldsymbol{\varepsilon} := (\varepsilon, \ldots, \varepsilon)$.

In each round $t = 1, 2, \ldots$, the agent chooses an arm $A_t$ and observes an independent draw $\mathbf{X}_t \sim \nu_{A_t}$ with $\mathbb{E}(\mathbf{X}_{A_t}) = \boldsymbol{\mu}_{A_t}$. We denote by $\mathbb{P}_\nu$ the law of the stochastic process $(\mathbf{X}_t)_{t \geq 1}$ and by $\mathbb{E}_\nu$, the

---

[2] A random variable $X$ is $\sigma-$subgaussian if for any $\lambda \in \mathbb{R}$, $\mathbb{E}(\exp(\lambda(X - \mathbb{E}(X)))) \leq \exp(\frac{\lambda^2 \sigma^2}{2})$.

expectation under $\mathbb{P}_\nu$. Let $\mathcal{F}_t := \sigma(A_1, \mathbf{X}_1, \ldots, A_t, \mathbf{X}_t)$ the $\sigma$-algebra representing the history of the process. An algorithm for PSI consists in : i) a *sampling rule* which determines which arm to sample at time $t$ based on history up to time $t - 1$, ii) a *stopping rule* $\tau$ which is a stopping time w.r.t the filtration $(\mathcal{F}_t)_{t \geq 1}$ and iii) a *recommendation rule* which is a $\mathcal{F}_\tau$-measurable random set $\widehat{S}_\tau$ representing the guess of the learner. The goal of the learner is to make a correct guess with high probability, using as few samples $\tau$ as possible. Before formalizing this, we introduce the different notion of correctness considered in this work, depending on parameters $\varepsilon_1 \geq, \varepsilon_2 \geq 0$ and $k \in [K]$. Our first criterion is the one considered by [4].

**Definition 2.** $\widehat{S} \subset \mathbb{A}$ *is correct for* $\varepsilon_1$-*PSI if* $\mathcal{S}^\star \subset \widehat{S} \subset \mathcal{S}^\star_{\varepsilon_1}$.

To introduce our second criterion, we need the following definition.

**Definition 3.** *Let* $\varepsilon_1, \varepsilon_2 \geq 0$. *A subset* $S \subset \mathbb{A}$ *is an* $(\varepsilon_1, \varepsilon_2)$-*cover of the* Pareto set *if* : $S \subset \mathcal{S}^\star_{\varepsilon_1}$ *and for any* $i \notin S$ *either* $i \notin \mathcal{S}^\star$ *or* $\exists j \in S$ *such that* $\boldsymbol{\mu}_i \prec \boldsymbol{\mu}_j + \varepsilon_2$.

The $\varepsilon$-accurate set of [27] is a particular case of $(\varepsilon_1, \varepsilon_2)$-cover for which $\varepsilon_1 = \varepsilon_2 = \varepsilon$. Allowing $\varepsilon_1 \neq \varepsilon_2$ generalizes the notion of $\varepsilon$-correct set and can be useful, e.g., in scenarios when we want to identify the exact Pareto set (setting $\varepsilon_1 = 0$) but allow some optimal arms to be discarded if they are too close (parameterized by $\varepsilon_2$) to another optimal arm already returned. We note however that the *sparse cover* of [4] is an $(\varepsilon, \varepsilon)$-cover with and additional condition that the arms in the returned set should not be too close to each over. Identifying a sparse cover from samples requires in particular to identify $\mathcal{S}^\star_{\varepsilon_1}$ hence it can not be seen as a relaxation of $\varepsilon_1$-PSI.

**Definition 4.** $\widehat{S} \subset \mathbb{A}$ *is correct for* $(\varepsilon_1, \varepsilon_2)$-*PSI if it is an* $(\varepsilon_1, \varepsilon_2)$-*cover of the Pareto set.*

**Definition 5.** $\widehat{S} \subset \mathbb{A}$ *is correct for* $\varepsilon_1$-*PSI-k if either* i) $|\widehat{S}| = k$ *and* $\widehat{S} \subset \mathcal{S}^\star_{\varepsilon_1}$ *or* ii) $|\widehat{S}| < k$ *and* $\mathcal{S}^\star \subset \widehat{S} \subset \mathcal{S}^\star_{\varepsilon_1}$ *holds.*

Given a specified objective ($\varepsilon_1$-PSI, $(\varepsilon_1, \varepsilon_2)$-PSI or $\varepsilon_1$-PSI-$k$), and a target risk parameter $\delta \in (0, 1)$, the goal of the agent is to build a $\delta$-correct algorithm, that is to guarantee that with probability larger than $1 - \delta$, her guess $\widehat{S}_\tau$ is correct for the given objective, while minimizing the number of samples $\tau$ needed to make the guess, called the *sample complexity*.

We now introduce two important quantities to characterize the (Pareto) optimality or sub-optimality of the arms. For any two arms $i, j$, we let

$$\mathrm{m}(i, j) := \min_{1 \leq d \leq D} (\mu_j^d - \mu_i^d), \text{ and } \mathrm{M}(i, j) := \max_{1 \leq d \leq D} (\mu_i^d - \mu_j^d),$$

which have the following interpretation. If $i \preceq j$, $\mathrm{m}(i, j)$ is the minimal quantity $\alpha \geq 0$ that should be added component-wise to $\boldsymbol{\mu}_i$ so that $\boldsymbol{\mu}_i + \boldsymbol{\alpha} \not\prec \boldsymbol{\mu}_j$, $\boldsymbol{\alpha} := (\alpha, \ldots, \alpha)$. Moreover, $\mathrm{m}(i, j) > 0$ if and only if $i \prec j$. Then, for any arms $i, j$, if $i \not\prec j$, $\mathrm{M}(i, j)$ is the minimum quantity $\alpha'$ such $\boldsymbol{\mu}_i \leq \boldsymbol{\mu}_j + \boldsymbol{\alpha'}$, $\boldsymbol{\alpha'} := (\alpha', \ldots, \alpha')$. We remark that $\mathrm{M}(i, j) < 0$ if and only if $i \prec j$. Our algorithms, presented in the next section, rely on confidence intervals on these quantities.

# 3 Adaptive Pareto Exploration

We describe in this section our sampling rule, Adaptive Pareto Exploration, and present three stopping and recommendation rules to which it can be combined to solve each of the proposed relaxation. Let $T_k(t) := \sum_{s=1}^{t-1} \mathbb{1}(A_s = k)$ be the number of times arm $k$ has been pulled up to round $t$ and $\widehat{\boldsymbol{\mu}}_k(t) := T_k(t)^{-1} \sum_{s=1}^{T_k(t)} \mathbf{X}_{k,s}$ the empirical mean of this arm at time $t$, where $\mathbf{X}_{k,s}$ denotes the $s$-th observation drawn *i.i.d* from $\nu_k$. For any arms $i, j \in \mathbb{A}$, we let

$$\mathrm{m}(i, j, t) := \min_d (\widehat{\mu}_j^d(t) - \widehat{\mu}_i^d(t)) \text{ and } \mathrm{M}(i, j, t) := \max_d (\widehat{\mu}_i^d(t) - \mu_j^d(t)).$$

The empirical Pareto set is defined as

$$\begin{aligned} S(t) &:= \{i \in \mathbb{A} : \nexists j \in \mathbb{A} : \widehat{\boldsymbol{\mu}}_i(t) \prec \widehat{\boldsymbol{\mu}}_j(t)\}, \\ &= \{i \in \mathbb{A} : \forall j \in \mathbb{A} \backslash \{i\}, \mathrm{M}(i, j, t) > 0\}. \end{aligned}$$

### 3.1 Generic algorithm(s)

Adaptive Pareto Exploration relies on a *lower/upper confidence bound* approach, similar to single-objective BAI algorithms like UGapEc [9], LUCB[15] and LUCB++ [26]. These three algorithms identify in each round two contentious arms: $b_t$: a current guess for the optimal arm (defined as the empirical best arm or the arm with the smallest upper bound on its sub-optimality gap), $c_t$: a contender of this arm; the arm which is the most likely to outperform $b_t$ (in all three algorithms, it is the arm with the largest upper confidence bound in $[K]\backslash\{b_t\}$). Then, either both arms are pulled (LUCB, LUCB++) or the least explored among $b_t$ and $c_t$ is pulled (UGapEc). The originality of our sampling rule lies in how to appropriately define $b_t$ and $c_t$ for the multi-objective setting. To define those, we suppose that there exists confidence intervals $[L_{i,j}^d(t,\delta), U_{i,j}^d(t,\delta)]$ on the difference of expected values for each pair of arms $(i,j)$ and each objective $d \in D$, such that introducing

$$\mathcal{E}_t := \bigcap_{i=1}^{K}\bigcap_{j\neq i}\bigcap_{d=1}^{D} \left\{L_{i,j}^d(t,\delta) \leq \mu_i^d - \mu_j^d \leq U_{i,j}^d(t,\delta)\right\} \text{ and } \mathcal{E} = \bigcap_{t=1}^{\infty}\mathcal{E}_t, \tag{1}$$

we have $\mathbb{P}(\mathcal{E}) \geq 1-\delta$. Concrete choices of these confidence intervals will be discussed in Section 3.2.

To ease the notation, we drop the dependency in $\delta$ in the confidence intervals and further define

$$M^-(i,j,t) := \max_d L_{i,j}^d(t) \quad \text{and} \quad M^+(i,j,t) := \max_d U_{i,j}^d(t) \tag{2}$$

$$m^-(i,j,t) := -M^+(i,j,t) \quad \text{and} \quad m^+(i,j,t) := -M^-(i,j,t). \tag{3}$$

**Lemma 1.** *For any round $t \geq 1$, if $\mathcal{E}_t$ holds, then for any $i,j \in \mathbb{A}$, $M^-(i,j,t) \leq M(i,j) \leq M^+(i,j,t)$ and $m^-(i,j,t) \leq m(i,j) \leq m^+(i,j,t)$.*

Noting that $\mathcal{S}_{\varepsilon_1}^\star = \{i \in \mathbb{A} : \forall j \neq i, M(i,j) + \varepsilon_1 > 0\}$, we define the following set of arms that are likely to be $\varepsilon_1$-Pareto optimal:

$$\text{OPT}^{\varepsilon_1}(t) := \{i \in \mathbb{A} : \forall j \in \mathbb{A}\backslash\{i\}, M^-(i,j,t) + \varepsilon_1 > 0\}.$$

**Sampling rule**  In round $t$, Adaptive Pareto Exploration samples $a_t$, the least pulled arm among two candidate arms $b_t$ and $c_t$ given by

$$b_t := \operatorname*{argmax}_{i \in \mathbb{A}\backslash\text{OPT}^{\varepsilon_1}(t)} \min_{j\neq i} M^+(i,j,t),$$

$$c_t := \operatorname*{argmin}_{j\neq b_t} M^-(b_t,j,t)$$

The intuition for their definition is the following. Letting $i$ be a fixed arm, note that $M(i,j) > 0$ for some $j$, if and only if there exists a component $d$ such that $\mu_i^d > \mu_j^d$ i.e $i$ is not dominated by $j$. Moreover, the larger $M(i,j)$, the more $i$ is non-dominated by $j$ in the sense that there exists $d$ such that $\mu_i^d \gg \mu_j^d$. Therefore, $i$ is strictly optimal if and only if for all $j \neq i$, $M(i,j) > 0$ i.e $\alpha_i := \min_{j\neq i} M(i,j) > 0$. And the larger $\alpha_i$, the more $i$ looks optimal in the sense that for each arm $j \neq i$, there exists a component $d_j$ for which $i$ is way better than $j$. As the $\alpha_i$ are unknown, we define $b_t$ as the maximizer of an optimistic estimate of the $\alpha_i$'s. We further restrict the maximization to arms for which we are not already convinced that they are optimal (by Lemma 1, the arms in $\text{OPT}^{\varepsilon_1}(t)$ are (nearly) Pareto optimal on the event $\mathcal{E}$). Then, we note that for a fixed arm $i$, $M(i,j) < 0$ if and only if $i$ is strictly dominated by $j$. And the smaller $M(i,j)$, the more $j$ is close to dominate $i$ (or largely dominates it): for any component $d$, $\mu_i^d - \mu_j^d$ is small (or negative). Thus, for a fixed arm $i$, $\operatorname{argmin}_{j\neq i} M(i,j)$ can be seen as the arm which is the closest to dominate $i$ (or which dominates it by the largest margin). By minimizing a lower confidence bound on the unknown quantity $M(b_t,j)$, our contender $c_t$ can be interpreted as the arm which is the most likely to be (close to) dominating $b_t$. Gathering information on both $b_t$ and $c_t$ can be useful to check whether $b_t$ can indeed be optimal.

Interestingly, we show in Appendix E that for $D = 1$, our sampling rule is close but not identical to the sampling rules used by existing confidence-based best arm identification algorithms.

| | Stopping condition | Recommendation | Objective |
|---|---|---|---|
| $\tau_{\varepsilon_1}$ | $Z_1^{\varepsilon_1}(t) > 0 \ \wedge Z_2^{\varepsilon_1}(t) > 0$ | $\mathcal{O}(\tau_{\varepsilon_1})$ | $\varepsilon_1$-PSI |
| $\tau_{\varepsilon_1,\varepsilon_2}$ | $Z_1^{\varepsilon_1,\varepsilon_2}(t) > 0 \ \wedge Z_2^{\varepsilon_1,\varepsilon_2}(t) > 0$ | $\text{OPT}^{\varepsilon_1}(\tau_{\varepsilon_1,\varepsilon_2})$ | $(\varepsilon_1,\varepsilon_2)$-PSI |
| $\tau^k$ | $|\text{OPT}^{\varepsilon_1}(t)| \geq k$ | $\text{OPT}^{\varepsilon_1}(\tau^k)$ | $\varepsilon_1$-PSI-$k$ |

Table 1: Stopping conditions and associated recommendation

**Stopping and recommendation rule(s)** Depending on the objective, Adaptive Pareto Exploration can be plugged in with different stopping rules, that are summarized in Table 1 with their associated recommendations. To define those, we define for all $i \in \mathbb{A}$, $\varepsilon_1, \varepsilon_2 \geq 0$,

$$g_i^{\varepsilon_2}(t) := \max_{j \neq i} \text{m}^-(i,j,t) + \varepsilon_2 \mathbb{1}\{j \in \text{OPT}^{\varepsilon_1}(t)\} \quad \text{and} \quad h_i^{\varepsilon_1}(t) := \min_{j \neq i} \text{M}^-(i,j,t) + \varepsilon_1.$$

and let $g_i(t) := g_i^0(t)$. Introducing

$$Z_1^{\varepsilon_1}(t) := \min_{i \in S(t)} h_i^{\varepsilon_1}(t), \ \text{ and } \ Z_2^{\varepsilon_1}(t) := \min_{i \in S(t)^c} \max(g_i(t), h_i^{\varepsilon_1}(t)),$$

for $\varepsilon_1$-PSI, our stopping rule is $\tau_{\varepsilon_1} := \inf\{t \geq K : Z_1^{\varepsilon_1}(t) > 0 \wedge Z_2^{\varepsilon_1}(t) > 0\}$ and the associated recommendation is $\mathcal{O}(\tau_{\varepsilon_1})$ where

$$\mathcal{O}(t) := S(t) \cup \{i \in S(t)^c : \nexists j \neq i : \text{m}^-(i,j,t) > 0\}$$

consists of the current empirical Pareto set plus some additional arms that have not yet been formally identified as sub-optimal. Those arms should be $(\varepsilon_1)$-Pareto optimal.

For $(\varepsilon_1,\varepsilon_2)$-PSI we define a similar stopping rule $\tau_{\varepsilon_1,\varepsilon_2}$ where the stopping statistics are respectively replaced with

$$Z_1^{\varepsilon_1,\varepsilon_2}(t) := \min_{i \in S(t)} \max(g_i^{\varepsilon_2}(t), h_i^{\varepsilon_1}(t)) \ \text{ and } Z_2^{\varepsilon_1,\varepsilon_2}(t) := \min_{i \in S(t)^c} \max(g_i^{\varepsilon_2}(t), h_i^{\varepsilon_1}(t))$$

with the convention $\min_\emptyset = +\infty$, and the recommendation is $\text{OPT}^{\varepsilon_1}(\tau_{\varepsilon_1,\varepsilon_2})$.

To tackle the $\varepsilon_1$-PSI-$k$ relaxation, we propose to couple $\tau_{\varepsilon_1}$ with an additional stopping condition checking whether $\text{OPT}^{\varepsilon_1}(t)$ already contains $k$ arms. That is, we stop at $\tau_{\varepsilon_1}^k := \min\left(\tau_{\varepsilon_1}, \tau^k\right)$ where $\tau^k := \inf\{t \geq K : |\text{OPT}^{\varepsilon_1}(t)| \geq k\}$ with associated recommendation $\text{OPT}^{\varepsilon_1}(\tau^k)$. Depending of the reason for stopping ($\tau_{\varepsilon_1}$ or $\tau^k$), we follow the corresponding recommendation.

**Lemma 2.** *Assume $\mathcal{E}$ holds. For $\varepsilon_1$-PSI (resp. $(\varepsilon_1,\varepsilon_2)$-PSI , $\varepsilon_1$-PSI-$k$), Adaptive Pareto Exploration combined with the stopping rule $\tau_{\varepsilon_1}$ (resp. $\tau_{\varepsilon_1,\varepsilon_2}$, resp. $\tau_{\varepsilon_1}^k$) outputs a correct subset.*

**Remark 1.** *We decoupled the presentation of the sampling rule to that of the "sequential testing" aspect (stopping and recommendation). We could even go further and observe that multiple tests could actually be run in parallel, for free. If we collect samples with APE (which only depends on $\varepsilon_1$), whenever one of the three stopping conditions given in Table 1 triggers, for any values of $\varepsilon_2$ or $k$, we can decide to stop and make the corresponding recommendation or continue and wait for another "more interesting" stopping condition to be satisfied. If $\mathcal{E}$ holds, a recommendation made at any such time will be correct for the objective associated to the stopping criterion (third column in Table 1).*

### 3.2 Our instantiation

We propose to instantiate the algorithms with confidence interval on the difference of pair of arms. For any pair $i, j \in \mathbb{A}$, we define a function $\beta_{i,j}$ such that for any $d \in [D]$, $U_{i,j}^d(t) = \widehat{\mu}_i^d(t) - \widehat{\mu}_j^d(t) + \beta_{i,j}(t)$ and $L_{i,j}^d(t) = \widehat{\mu}_i^d(t) - \widehat{\mu}_j^d(t) - \beta_{i,j}(t)$. We take from [20] the following confidence bonus for time-uniform concentration:

$$\beta_{i,j}(t) := 2\sqrt{\left(C^g\left(\frac{\log\left(\frac{K_1}{\delta}\right)}{2}\right) + \sum_{a \in \{i,j\}} \log(4 + \log(T_a(t)))\right)\left(\sum_{a \in \{i,j\}} \frac{1}{T_a(t)}\right)}, \quad (4)$$

where $K_1 := K(K-1)D/2$ and $C^g \approx x + \log(x)$ is a calibration function. They result in the simple expressions $\text{M}^\pm(i,j,t) = \text{M}(i,j,t) \pm \beta_{i,j}(t)$ and $\text{m}^\pm(i,j,t) = \text{m}(i,j,t) \pm \beta_{i,j}(t)$. As an

**Algorithm 1:** $\varepsilon_1$-APE-$k$: Adaptive Pareto Exploration for $\varepsilon_1$-PSI-$k$

---

**Data:** parameter $\varepsilon_1 \geq 0$, $k \in [K]$
**initialize :** sample each arm once, set $t = K$, $T_i(K) = 1$ for any $i \in \mathbb{A}$
**for** $t = K+1, \ldots,$ **do**

    $S(t) = \{i \in \mathbb{A} : \forall j \in \mathbb{A}\backslash\{i\}, \mathrm{M}(i,j,t) > 0\}$;
    $\mathrm{OPT}^{\varepsilon_1}(t) = \{i \in \mathbb{A} : \forall j \in \mathbb{A}\backslash\{i\}, \mathrm{M}(i,j,t) - \beta_{i,j}(t) + \varepsilon_1 > 0\}$;
    **if** $|\mathrm{OPT}^{\varepsilon_1}(t)| \geq k$ **then**
        $\lfloor$ **break** and output $\mathrm{OPT}^{\varepsilon_1}(t)$
    **if** $Z_1^{\varepsilon_1}(t) > 0 \ \wedge\ Z_2^{\varepsilon_1}(t) > 0$ **then**
        $\lfloor$ **break** and output $\mathcal{O}(t) = S(t) \bigcup \{i \in S(t)^c : \nexists j \neq i : \mathrm{m}(i,j,t) - \beta_{i,j}(t) > 0\}$
    $b_t := \mathrm{argmax}_{i \in \mathbb{A}\backslash\mathrm{OPT}^{\varepsilon_1}(t)} \min_{j \neq i} [\mathrm{M}(i,j,t) + \beta_{i,j}(t)]$;
    $c_t := \mathrm{argmin}_{i \neq b_t} [\mathrm{M}(b_t, j, t) - \beta_{b_t, j}(t)]$;
    **sample** $a_t := \mathrm{argmin}_{i \in \{b_t, c_t\}} T_i(t)$;

---

example, we state in Algorithm 1 the pseudo-code of APE combined the stopping rule suited for the $k$-relaxation of $\varepsilon_1$-PSI, which we refer to as $\varepsilon_1$-APE-$k$.

In Appendix F, we also study a different instantiation based on confidence bounds of the form $U_{i,j}(t) = U_i(t) - L_j(t)$ where $[L_i(t), U_i(t)]$ is a confidence interval on $\mu_i$. This is the approach followed by LUCB for $D = 1$ and prior work on Pareto identification [4, 27]. In practice we advocate the use of the pairwise confidence intervals defined above, even if our current analysis does not allow to quantify their improvement. For the LUCB-like instantiation, we further derive in Appendix F an upper bound on the expected stopping time of APE for the different stopping rules.

## 4 Theoretical analysis

In this section, we state our main theorem on the sample complexity of our algorithms and give a sketch of its proof.

First let us introduce some quantities that are needed to state the theorem. The sample complexity of the algorithm proposed by [4] for $(\varepsilon_1)$-Pareto set identification scales as a sum over the arms $i$ of $1/(\Delta_i \vee \varepsilon_1)^2$ where $\Delta_i$ is called the sub-optimality gap of arm $i$ and is defined as follows. For a sub-optimal arm $i \notin \mathcal{S}^\star(\mathcal{X})$,

$$\Delta_i := \max_{j \in \mathcal{S}^\star} \mathrm{m}(i,j),$$

which is the smallest quantity that should be added component-wise to $\boldsymbol{\mu}_i$ to make $i$ appear Pareto optimal w.r.t $\{\boldsymbol{\mu}_i : i \in \mathbb{A}\}$. For a Pareto optimal arm $i \in \mathcal{S}^\star(\mathcal{X})$, the definition is more involved:

$$\Delta_i := \begin{cases} \min_{j \in \mathbb{A}\backslash\{i\}} \Delta_j & \text{if } \mathcal{S}^\star := \{i\} \\ \min(\delta_i^+, \delta_i^-) & \text{else,} \end{cases}$$

where

$$\delta_i^+ := \min_{j \in \mathcal{S}^\star\backslash\{i\}} \min(\mathrm{M}(i,j), \mathrm{M}(j,i)) \quad \text{and} \quad \delta_i^- := \min_{j \in \mathbb{A}\backslash\mathcal{S}^\star} \{(\mathrm{M}(j,i))^+ + \Delta_j\}.$$

For $x \in \mathbb{R}, (x)^+ := \max(x, 0)$. We also introduce some additional notion needed to express the contribution of the $k$-relaxation. Let $1 \leq k \leq K$. For any arm $i$, let $\omega_i = \min_{j \neq i} \mathrm{M}(i,j)$ and define

$$\omega^k := \overset{k}{\max_{i \in \mathbb{A}}}\ \omega_i, \qquad \mathcal{S}^{\star,k} := \overset{1 \ldots k}{\mathrm{argmax}_{i \in \mathbb{A}}}\ \omega_i,$$

with the $k$-th max and first to $k$-th argmax operators. Observe that $\omega^k > 0$ if and only if $|\mathcal{S}^\star(\mathcal{X})| \geq k$.

**Theorem 1.** *Fix a risk parameter $\delta \in (0,1)$, $\varepsilon_1 \geq 0$, let $k \leq K$ and $\nu$ a bandit with 1-subgaussian marginals. With probability at least $1 - \delta$, $\varepsilon_1$-APE-$k$ recommends a correct set for the $\varepsilon_1$-PSI-$k$ objective and stops after at most*

$$\sum_{a \in \mathbb{A}} \frac{88}{\widetilde{\Delta}_a^2} \log\left(\frac{2K(K-1)D}{\delta} \log\left(\frac{12e}{\widetilde{\Delta}_a}\right)\right),$$

*samples, where for each $a \in \mathbb{A}$, $\widetilde{\Delta}_a := \max(\Delta_a, \varepsilon_1, \omega^k)$.*

First, when $k = K$, observing that $\varepsilon_1$-APE-$K$ provides a $\delta$-correct algorithm for $\varepsilon_1$-PSI, our bound improves the result of [4] for the $\varepsilon_1$-PSI problem in terms of constant multiplicative factors and $\log \log \Delta^{-1}$ terms instead of $\log \Delta^{-2}$. It nearly matches the lower bound of Auer et al.[4] for the $\varepsilon_1$-PSI problem (Theorem 17 therein). It also shows the impact of the $k$-relaxation on the sample complexity. In particular, we can remark that for any arm $i \in \mathcal{S}^\star \backslash \mathcal{S}^{\star,k}$, $\max(\Delta_i, \omega_k) = \omega_k$. Intuitively, it says that we shouldn't pay more than the cost of identifying the $k$-th optimal arm, ordered by the $\omega_i'$s. A similar result has been obtained for the *any $k-$sized subset of the best $m$* problem [25]. But the authors have shown the relaxation only for the best $m$ arms while our result shows that even the sub-optimal arms should be sampled less.

In Appendix D, we prove the following lower bound showing that in some scenarios, $\varepsilon_1$-APE-$k$ is optimal for $\varepsilon_1$-PSI-$k$ , up to $D \log(K)$ and constant multiplicative terms. We note that for $\varepsilon_1$-PSI a lower bound featuring the gaps $\Delta_a$ and $\varepsilon_1$ was already derived by Auer et al. [4].

**Theorem 2.** *There exists a bandit instance $\nu$ with $|\mathcal{S}^\star| = p \geq 3$ such that for $k \in \{p, p-1, p-2\}$ any $\delta$-correct algorithm for 0-PSI-$k$ verifies*

$$\mathbb{E}_\nu(\tau_\delta) \geq \frac{1}{D} \log\left(\frac{1}{\delta}\right) \sum_{a=1}^{K} \frac{1}{(\Delta_a^k)^2},$$

*where $\Delta_a^k := \Delta_a + \omega^k$ and $\tau_\delta$ is the stopping time of the algorithm.*

In Appendix C, we prove that Theorem 1 without the $\omega^k$ terms also holds for $(\varepsilon_1, \varepsilon_2)$-APE. This does not justifies the reduction in sample complexity when setting $\varepsilon_2 > 0$ in $(\varepsilon_1, \varepsilon_2)$-PSI observed in our experiments but it at least guarantees that the $\varepsilon_2$-relaxation doesn't make things worse.

Furthermore, since our algorithm allows $\varepsilon_1 = 0$, it is also an algorithm for BAI when $D = 1, \varepsilon_1 = 0$. We prove in Appendix E that in this case, the gaps $\Delta_i's$ matches the classical gaps in BAI [2, 18] and we derive its sample complexity from Theorem 1 showing that it is similar in theory to UGap [9], LUCB[15] and LUCB++ [26] but have better empirical performance.

**Sketch of proof of Theorem 1**   Using Proposition 24 of [20] we first prove that the choice of $\beta_{i,j}$ in (4) yields $\mathbb{P}(\mathcal{E}) \geq 1 - \delta$ for the good event $\mathcal{E}$ defined in (1). Combining this result with Lemma 2, yields that $\varepsilon_1$-APE-$k$ is correct with probability at least $1 - \delta$.

The idea of the remaining proof is to show that under the event $\mathcal{E}$, if APE has not stopped at the end of round $t$, then the selected arm $a_t$ has not been explored enough. The first lemma showing this is specific to the stopping rule $\tau_{\varepsilon_1}^k$ used for $\varepsilon_1$-PSI-$k$.

**Lemma 3.** *Let $\varepsilon_1 \geq$ and $k \in [K]$. If $\mathcal{E}_t$ holds and $t < \tau_{\varepsilon_1}^k$ then $\omega^k \leq 2\beta_{a_t,a_t}(t)$.*

The next two lemmas are more general as they apply to different stopping rules.

**Lemma 4.** *Let $\varepsilon_1 \geq 0$. Let $\tau = \tau_{\varepsilon_1}^k$ for some $k \in [K]$ or $\tau = \tau_{\varepsilon_1,\varepsilon_2}$ for some $\varepsilon_2 \geq 0$. If $\mathcal{E}_t$ holds and $t < \tau$ then $\Delta_{a_t} \leq 2\beta_{a_t,a_t}(t)$.*

**Lemma 5.** *Let $\varepsilon_1 \geq 0$ and $\tau$ be as in Lemma 4. If $\mathcal{E}_t$ holds and $t < \tau$ then $\varepsilon_1 \leq 2\beta_{a_t,a_t}(t)$.*

As can be seen in Appendix B, the proofs of these three lemmas heavily rely on the specific definition of $b_t$ and $c_t$. In particular, to prove Lemma 4 and 5, we first establish that when $t < \tau$, any arm $j \in \mathbb{A}$ satisfies $\mathrm{m}(b_t, j, t) \leq \beta_{b_t,j}(t)$. The most sophisticated proof is then that of Lemma 4, which relies on a case distinction based on whether $b_t$ or $c_t$ belongs to the set of optimal arms.

These lemmas permit to show that, on $\mathcal{E}_t$ if $t < \tau_{\varepsilon_1}^k$ then $\widetilde{\Delta}_{a_t} < 2\beta_{a_t,a_t}(t) \leq 2\beta^{T_{a_t}(t)}$, where we define $\beta^n$ to be the expression of $\beta_{i,j}(t)$ when $T_i(t) = T_j(t) = n$. Then we have

$$
\begin{aligned}
\tau_{\varepsilon_1}^k \mathbb{1}\{\mathcal{E}\} &\leq \sum_{t=1}^{\infty} \sum_{a \in \mathbb{A}} \mathbb{1}\left\{\{a_t = a\} \wedge \left\{\widetilde{\Delta}_a \leq 2\beta_a^{T_a(t)}\right\}\right\} \\
&\leq \sum_{a \in \mathbb{A}} \inf\left\{n \geq 2 : \widetilde{\Delta}_a > 2\beta^n\right\}.
\end{aligned}
\tag{5}
$$

A careful upper-bounding of the RHS of (5) completes the proof of Theorem 1.

$\square$

# 5 Experiments

We evaluate the performance of Adaptive Pareto Exploration on a real-world scenario and on synthetic random Bernoulli instances. For a fair comparison, Algorithm 1 of [4], referred to as PSI-Unif-Elim and APE are both run with our confidence bonuses $\beta_{i,j}(t)$ on pairs of arms, which considerably improve single-arm confidence bonuses[3]. As anytime confidence bounds are known to be conservative, we use $K_1 = 1$ in (4) instead of its theoretical value coming from a union bound. Still, in all our experiments, the empirical error probability was (significantly) smaller than the target $\delta = 0.1$.

**Real-world dataset**    COV-BOOST [24] is phase 2 trial which was conducted on 2883 participants to measure the immunogenicity of different Covid-19 vaccines as third dose (booster) in various combinations of initially received vaccines (first two doses). This resulted in a total of 20 vaccination strategies being assessed, each of them defined by the vaccines received as first, second and third dose. The authors have reported the average responses induced by each candidate strategy on cohorts of participants, measuring several immunogenicity markers. From this study, we extract and process the average response of each strategy to 3 specific immunogenicity indicators: two markers of antibody response and one of the cellular response. The outcomes are assumed to have a log-normal distribution [24]. We use the average (log) outcomes and their variances to simulate a multivariate Gaussian bandit with $K = 20, D = 3$. We give in Appendix H.2 some additional details about the processing of the data, and report the means and variances of each arm. In Appendix H.1 we further explain how APE can be simply adapted when the marginals distributions of the arms have different variances. In this experiment, we set $\varepsilon_1 = 0, \delta = 0.1$ and compare PSI-Unif-Elim to 0-APE-$k$ (called

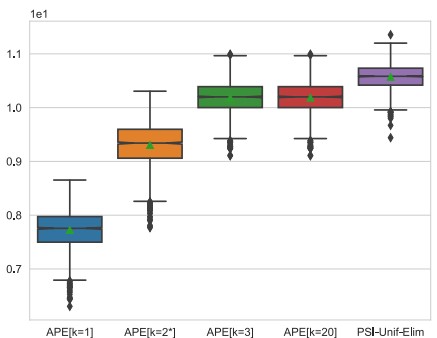 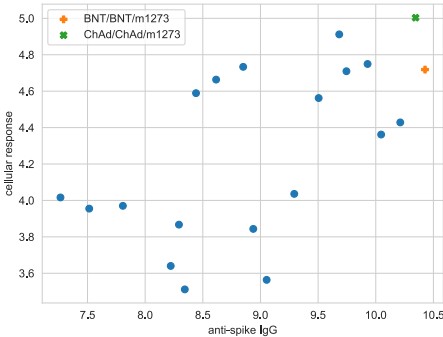

Figure 1: On the left is the log of the empirical sample complexity of PSI-Unif-Elim and APE on the real-world scenario plot (right) for 2 out of the 3 immunogenicity indicators.

APE-$k$ in the sequel) for different values of $k$. The empirical distribution of the sample complexity of the algorithms, averaged over 2000 independent runs, are reported in Figure 1. The results are shown in log-scale (y-axis is the log of the sample complexity) to fit in the same figure. As $|\mathcal{S}^\star| = 2$, we first observe that, without the relaxation (i.e. for $k > 3$), APE outperforms its state-of-the-art competitor PSI-Unif-Elim . Moreover for $k = 1$ or $k = 2$, the sample complexity of APE-$k$ is significantly reduced. For $k = 2$ when the stopping time $\tau^k$ is reached some sub-optimal arms have possibly not yet been identified as such, while for $k = 3$, even if the optimal arms have been identified, the remaining arms have to be sampled enough to ensure that they are sub-optimal before stopping. This explains the gap in sample complexity between $k = 2$ and $k = 3$. In Appendix H.3, we compare APE to an adaptation of PSI-Unif-Elim for the $k$-relaxation, showing that APE is always preferable.

**Experiments on random instances**    To further illustrate the impact of the $k$-relaxation and to strengthen the comparison with PSI-Unif-Elim , we ran the previous algorithms on 2000 randomly generated multi-variate Bernoulli instances, with $K = 5$ arms and different values of the dimension $D$. We set $\delta = 0.1$ and $\varepsilon_1 = 0.005$ (to have reasonable running time). The averaged sample complexities are reported in Table 2. We observe that APE (with $k = K$) uses 20 to 25% less samples

---

[3]In their experiments, [4] already proposed the heuristic use of confidence bonuses of this form

than PSI-Unif-Elim and tends to be more efficient as the dimension increases (and likely the size of the Pareto set, since the instances are randomly generated). We also note that identifying a $k$-sized subset of the Pareto set requires considerably less samples than exact PSI. In Appendix H.3 we also provide examples of instances for which APE takes up to 3 times less samples than PSI-Unif-Elim .

|  | $\varepsilon_1$-APE-1 | $\varepsilon_1$-APE-2 | $\varepsilon_1$-APE-3 | $\varepsilon_1$-APE-4 | $\varepsilon_1$-APE-5 | $\varepsilon_1$-PSI-Unif-Elim |
|---|---|---|---|---|---|---|
| $D = 2$ | 811 | 39530 | 109020 | 145777 | 150844 | 190625 |
| $D = 4$ | 214 | 6410 | 19908 | 68061 | 124001 | 157584 |
| $D = 8$ | 119 | 204 | 405 | 1448 | 20336 | 27270 |

Table 2: Average sample complexity over 2000 random Bernoulli instances with $K = 5$ arms. On average the size of the Pareto set was $(2.295, 4.0625, 4.931)$ respectively for the dimensions $2, 4, 8$.

To illustrate the impact of the $\varepsilon_2$ relaxation, setting $\varepsilon_1 = 0$ we report the sample complexity of APE associated with the stopping time $\tau_{0,\varepsilon_2}$ for 20 equally spaced values of $\varepsilon_2 \in [0.01, 0.05]$, averaged over 2000 random Bernoulli instances. Figure 2 shows the decrease of the average sample complexity when $\varepsilon_2$ increases (left) and the average ratio of the size of the returned set to the size of the Pareto set (right). Note that for $\varepsilon_1 = 0$, we have $\mathcal{O}(\tau_{0,\varepsilon_2}) \subset \mathcal{S}^\star$. The average sample complexity reported decreases up to $86\%$ for the instance with $K = 5, D = 2$ while the returned set contains more than $90\%$ of the Pareto optimal arms. In Appendix H.3, we further illustrate the behavior of APE with the $\varepsilon_2$ relaxation on a fixed instance in dimension 2.

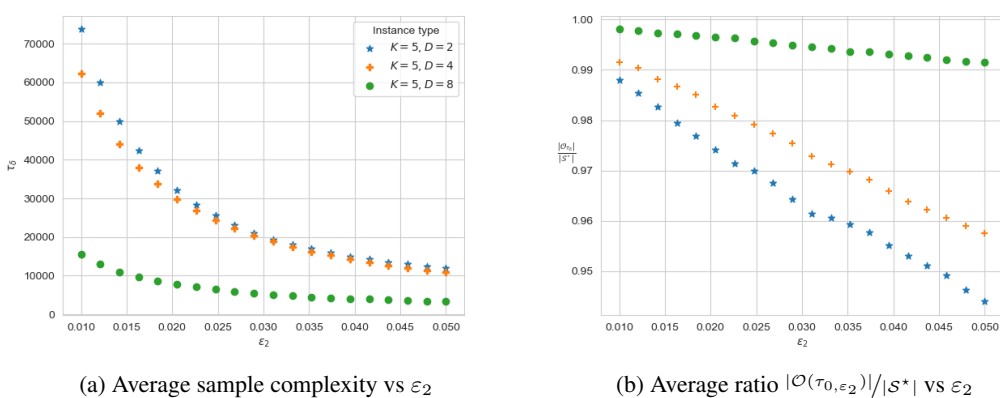

(a) Average sample complexity vs $\varepsilon_2$        (b) Average ratio $|\mathcal{O}(\tau_{0,\varepsilon_2})|/|\mathcal{S}^\star|$ vs $\varepsilon_2$

Figure 2: APE with $\tau_{0,\varepsilon_2}$ averaged over 2000 random Bernoulli instance with $K = 5$ arms.

## 6  Conclusion and perspective

We proposed and analyzed APE, an adaptive sampling rule in multi-variate bandits models that when coupled with different stopping rules can tackle different relaxations of the fixed-confidence Pareto Set Identification problem. Our experiments revealed the good performance of the resulting algorithms compared to the state-of-the-art PSI algorithm as well as the great reductions in sample complexity brought by the relaxations. In future work, we intend to make our algorithms more practical for possible applications to clinical trials. For this purpose, as measuring efficacy takes time, we will investigate its adaptation to a batch setting, following, e.g. the work of [13] for BAI. We will also investigate the use of APE beyond the fixed-confidence setting, to the possibly more realistic fixed-budget [2] or anytime exploration [14] settings. To the best of our knowledge, no algorithm exists in the literature for PSI in such settings. Finally, following the works of [28, 4], we defined the $\varepsilon_1, \varepsilon_2$ relaxations with scalar values, so that the same slack applies to all components. Although we could easily modify our algorithms to tackle vectorial values $\boldsymbol{\varepsilon}_1, \boldsymbol{\varepsilon}_2$, so far we could only prove a dependence on $\min_d \varepsilon_1^d$ in the sample complexity. We intend to study the right quantification in the sample complexity when $\boldsymbol{\varepsilon}_1$ and $\boldsymbol{\varepsilon}_2$ are vectorial.

## Acknowledgments and Disclosure of Funding

Cyrille Kone is funded by an Inria/Inserm PhD grant. Emilie Kaufmann acknowledges the support of the French National Research Agency under the projects BOLD (ANR-19-CE23-0026-04) and FATE (ANR22-CE23-0016-01).

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
