# OpenReview forum: "Adaptive Algorithms for Relaxed Pareto Set Identification"
_NeurIPS.cc/2023/Conference — NeurIPS 2023 poster_

### Official Review · Reviewer_rUsc · 2023-07-01

**Soundness:** 3 good
**Presentation:** 3 good
**Contribution:** 3 good
**Rating:** 6
**Confidence:** 2

**Summary:**

This paper studies a less common setting that arms are multi-variate distributions. In the expiration period, the agent seeks to identify the arm which is Pareto optimal.

This problem is formulated as a fixed-confidence identification problem. The most novel setting in that, the fixed confidence identification problem s extended for multi-objective settings.

This paper proposed a novel sampling rule called Adaptive Pareto Exploration. The central idea is to identify the two contentious arms and sample both or one of them.

The paper also gives the theoretical guarantee that  the proposed method can recommend a correct set whp after most a given number of iterations.





**Strengths:**

This paper proposes a principled method and provides theoretical guarantees.


**Weaknesses:**

It seems trival to me to extend the single objective best arm  identification problem to the mo setting.

**Questions:**

1. Can the proposed method identify the full (relaxed) Pareto front?



**Limitations:**

No.

---

> ### Author Rebuttal · Authors · 2023-08-08
>
> Thank you for your review.
>
> * We would like to clarify from your summary that our algorithm is not meant to identify "the arm which is Pareto optimal" as there might be more than one Pareto optimal arm. We propose a sampling rule called Adaptive Pareto Exploration that when combined with different stopping rules, can solve different objectives: identify the entire Pareto set ($0$-PSI in our terminology), identify the entire Pareto set and possibly a few extra arms close to it ($\varepsilon_1$-PSI), identify at most $k$ arms that are (nearly) Pareto optimal ($\varepsilon_1$-PSI-$k$) or identify a "representative" subset of the Pareto set ($(\varepsilon_1,\varepsilon_2)$-PSI). We hope that this clarifies our contribution and answers your question.
>
> * We strongly disagree that it is trivial to extend the single objective best arm identification problem to the mo setting. While PSI (Pareto set identification) is a natural extension of the best arm identification problem in standard bandits, PSI is actually a more challenging problem. In PSI we don't know the number of optimal arms beforehand: it can go from $1$ to $K$, while in BAI the classical assumption is that there is a single optimal arm. When relaxations are further taken into account, there are multiple valid solutions (e.g. for $\varepsilon_1$-PSI-$k$ when the size of the Pareto set is larger than $k$, any subset of size $k$ is a valid guess), which creates some additional difficulties even in the uni-dimensional setting [10]. It is true that our sampling rule shares a common structure with that of existing adaptive BAI algorithms which rely on playing two contentious arms ($b_t$ and $c_t$ in our paper), often the empirical best arm  and a contender (LUCB, UGap or more recently Top Two algorithms). However, it actually took us several iterations to come up with the right definitions for $b_t$ and $c_t$ for the PSI problem, which interestingly do not exactly coincide with the choices in prior BAI algorithms when $D=1$ (see Appendix E). You can also read our answer to reviewer 9Ln4 for a better intuition about our definition of $b_t$ and $c_t$, which will be included in our revision. As a start, in the multi-objective case we no longer have an ``empirical best arm" but a set of arms of random size that could equally be optimal. Likewise, many results and techniques known so far for BAI do not directly apply to our setting and our analysis required specific results to pin down the behavior of our algorithm: Lemma 3 and Lemma 4. We will be happy to sketch their proof (and their crucial ingredient Lemma 8 currently stated in appendix) if we are given an extra page for our revision.

---

> > ### Comment · Reviewer_rUsc · 2023-08-17
> > **reply**
> >
> > I have read the response. And since I am not an expert in this field. I am not confident with my score.

---

### Official Review · Reviewer_yuHu · 2023-07-07

**Soundness:** 4 excellent
**Presentation:** 4 excellent
**Contribution:** 3 good
**Rating:** 6
**Confidence:** 3

**Summary:**

The primary objective of the paper is to address the challenge of identifying a set of arms, consisting of at most $k$ arms, where each arm is either Pareto optimal or close to Pareto optimal. The paper explains the concept of Pareto optimality within the context of bandit problems. To tackle this problem, the paper presents the $\epsilon_1$-APE-$k$ (Adaptive Pareto Exploration) algorithm and establishes an upper bound on the sample complexity. Furthermore, empirical evidence is provided to demonstrate the effectiveness of the proposed algorithm.

**Strengths:**

One strength of the paper is its focus on addressing the problem of Pareto optimality in bandit literature, which is an underexplored area. The paper introduces a novel problem setup and clearly defines the goals for different scenarios related to Pareto optimal identification, including the identification of Pareto optimal sets, near Pareto optimal sets, and at-most $k$ near Pareto optimal sets.

The paper excels in providing a comprehensive discussion on sample complexity upper bounds, highlighting the theoretical aspects of the problem. It also establishes connections to the existing literature on Pareto optimality in bandits, demonstrating a solid understanding of the research landscape.

Furthermore, the paper offers ample experimental evidence to support its claims, including experiments conducted on real-world datasets. This empirical validation strengthens the credibility of the proposed approaches and enhances the practical relevance of the research.

**Weaknesses:**

One weakness of the paper is the absence of references to real-world applications that could benefit from the proposed framework. For instance, discussing potential applications such as A/B testing in clinical trials would enhance the practical relevance and broader impact of the research.

Another weakness is the lack of discussion of lower bounds in the main paper. Given that the problem setup for Pareto set identification is relatively new, it would be valuable to explore more properties of the lower bounds to gain insights into the tightness of the derived upper bounds. This would provide a more comprehensive understanding of the problem and its inherent complexities.

The paper lacks a thorough discussion on the computational complexity of the $\epsilon_1$-APE-$k$ algorithm. Considering the importance of computational efficiency in practical applications, it would be beneficial to address

**Questions:**

One potential weakness is the lack of information regarding scalability issues encountered when running large-scale experiments using the $\epsilon_1$-APE-$k$ algorithm. It would be valuable to understand whether the algorithm faces any challenges in handling larger datasets or more complex scenarios. Additionally, providing references on the scale of parameters used in A/B testing for applications beyond COVID datasets would further strengthen the practicality of the proposed approach.

Another point to consider is the limited discussion on the broader scope of utilizing Pareto sets in various applications apart from clinical trials. Exploring and discussing other potential domains where Pareto sets could be beneficial would enhance the paper's impact and shed light on additional practical applications.

**Limitations:**

No limitations or potential impact of their work discussed

---

> ### Author Rebuttal · Authors · 2023-08-08
>
> Thank you for your review. We address your concerns and answer your questions below.
>
> * We will add a paragraph about the computational complexity of $\varepsilon_1$-APE-$k$ in Section 5, with some details in Appendix H. The time complexity of $\varepsilon_1$-APE-$k$ is $\mathcal{O}(K^2D)$ and its memory complexity is $\mathcal{O}(K^2)$. The computational bottleneck is the computation of the $M(i,j,t)$ for each $(i,j) \in [K] \times [K]$, which requires a triple-nested for-loop over $[K]\times [K]\times[D]$). We have implemented the algorithm in C++17 compiled with GCC12. To give an idea of the run time, a single run on a random Bernoulli instance with $K=1000, D=10$ takes around 4 minutes for $0.1-$APE-$1000$ on a personal computer (a single 3GHz ARM core used, 8 GB RAM, 256 GB disk storage) with $\delta=0.01$ and no particular optimization. We are aware that due its fully sequential nature, our algorithm may have a higher computational cost compared to uniform sampling strategies which typically proceed in batches. However, in our implementation the most time-consuming operation was actually collecting a sample from the selected arm(s), especially for multivariate Gaussians. So that finally, in the experiments, our algorithm which ultimately requires less samples had in practice a smaller computational cost compared to PSI-Unif-Elim which uses uniform sampling.
>
> * We will provide in our revision more examples of practical cases in which adaptively identifying a Pareto set is meaningful. In the current version of the introduction we mostly focused on the motivating examples used in our experimental evaluation, for which the $k$-relaxation is especially relevant, referring only briefly to other examples contained in [5] (see l.42-44). We will elaborate on those and provide others. For example in [5] the authors have evaluated the PAL (Gaussian Process based) algorithm on the SW-LLVM dataset, which is a dataset of 1023 compiler settings and 11 objective indicators (memory footprint, performance etc.). Since it is unlikely that a single compiler setting optimizes all the 11 objectives, it is meaningful to find compiler settings that are Pareto optimal. Another application is hardware design optimization [5,7]. The idea is similar to software design but the arms are different hardware implementations designed to solve a given problem. The usual objectives for this application include chip area, throughput, energy consumption and runtime.
> Other applications include A/B/n testing for marketing or online recommender systems in which it is common to jointly optimize multiple (possibly conflicting) objectives such as user behavioral metrics (e.g. clicks, streams, dwell time, etc), supplier exposure objectives (e.g. diversity) and platform centric objectives (e.g. promotions). We will add a reference to the paper (Mehrotra, Xue, Lalmas. *Bandit based optimization of multiple objectives on a music streaming platform*, KDD 2020). Although we are not aware of the scale of the parameters used for these kind of applications, benchmarks of simple heuristics on undisclosed datasets in this paper have shown fair practical performance, which we are confident could be outperformed by our method. Going back to potential applications to adaptive clinical trials, we could also mention other examples besides vaccinology in which it is common to monitor several possibly conflicting objectives. For example one can think of clinical trials combining patient-reported outcomes (e.g. quality-of-life questionnaires) and clinical outcomes (e.g. tumour remission) or clinical trials in neurocognitive diseases that use a lot of different tests assessing different neurocognitive dimensions (ex. executive functions and memory tests).
>
> * Due to space limitation, we had to state our lower bound (Theorem 3) in the appendix, but we will move it to the main text if given some extra space for our revision, and also mention the existing lower bound of [6] (not for the $k$-relaxation). We emphasize that our lower bound is only a worse case result (in the spirit of the one derived by [11] for the problem of finding a $k$-sized subset of the top $m$ arms in a standard bandit). We prove that there exists some instances in which the sample complexity has to scale with our gaps. We will leave as an open question whether we can derive a tighter lower bound which is valid for every instance. But deriving such bounds in the presence of multiple correct outputs is known to be challenging [10].

---

> > ### Comment · Reviewer_yuHu · 2023-08-19
> >
> > I would like to thank the authors for their detailed responses, they helped me clear up some of my queries.

---

### Official Review · Reviewer_J7Kd · 2023-07-09

**Soundness:** 2 fair
**Presentation:** 2 fair
**Contribution:** 2 fair
**Rating:** 5
**Confidence:** 2

**Summary:**

This paper extends the best arm bandit problem to a multi-objective version to identify the arms that are in the Pareto front.

**Strengths:**

This problem is new and the author provides a clear practice motivation of this problem.

The theory study seems valid and complete.

**Weaknesses:**

The math notation seems a bit over-complicated and I was wondering can you simplify it?

Overall, extending the best arm bandit problem into multi-object setting seems straightforward, which limits the novelty of this paper.

The experiment result looks a bit hand-wavy. I understand that there might not be any algorithm specifically designed for such problem, but I was wondering how the proposed method compares with some naively modified baselines?



**Questions:**

In the definition of Pareto front, why we don't consider the variance of the distribution?

**Limitations:**

See above

---

> ### Author Rebuttal · Authors · 2023-08-08
>
> Thank you for your review. We comment about each weakness mentioned and answer your question below.
>
>  * Due to the multi-dimensional setting and the fact that we consider three relaxations simultaneously, we agree that the notation can be a bit heavy. We will double check if some simplification is possible.
>
>  * While PSI (Pareto set identification) is a natural extension of the best arm identification (BAI) problem in standard bandits, PSI is actually a more challenging problem. In PSI we don't know the number of optimal arms beforehand: it can go from $1$ to $K$, while in BAI the classical assumption is that there is a single optimal arm. When relaxations are further taken into account, there are multiple valid solutions (e.g. for $\varepsilon_1$-PSI-$k$ when the size of the Pareto set is larger than $k$, any subset of size $k$ is a valid guess), which creates some additional difficulties even in the uni-dimensional setting [10]. It is true that our sampling rule shares a common structure with that of existing adaptive BAI algorithms which rely on playing two contentious arms ($b_t$ and $c_t$ in our paper), often the empirical best arm and a contender (LUCB, UGap or more recently Top Two algorithms). However, it actually took us several iterations to come up with the right definitions for $b_t$ and $c_t$ for the PSI problem, which interestingly do not exactly coincide with the choices in prior BAI algorithms when $D=1$ (see Appendix E). You can also read our answer to reviewer 9Ln4 for a better intuition about our definition of $b_t$ and $c_t$, which will be included in our revision. As a start, in the multi-objective case we no longer have an ``empirical best arm" but a set of arms of random size that could equally be optimal. Likewise, many results and techniques known so far for BAI do not directly apply to our setting and our analysis required specific results to pin down the behavior of our algorithm: Lemma 3 and Lemma 4. We will be happy to sketch their proof (and their crucial ingredient Lemma 8 currently stated in appendix) if we are given an extra page for our revision.
>
> * The experiments reported in the main paper are mostly meant to illustrate the reduction in sample complexity obtained when solving the $\varepsilon_1$-PSI-$k$ relaxation (for different values of $k$), compared to the state-of-the-art algorithm for the (unrelaxed) $\varepsilon_1$-PSI problem from [6], which we call PSI-Unif-Elim. In particular, when $k=K$ (the total number of arms), we are comparing two algorithms ($\varepsilon_1$-APE-$K$ and PSI-Unif-Elim) designed for the same objective: $\varepsilon_1$-PSI, showing that our proposal leads to a reduced sample complexity (by 25$\\%$ on average, Figure 1 and Table 2 of the main paper). In Appendix H.3.3. we further propose a naive modification of PSI-Unif-Elim for the $\varepsilon_1$-PSI-$k$ objective (Algorithm 3) and compare it to $\varepsilon_1$-APE-$k$. The results, reported in Figure 8 show that $\varepsilon_1$-APE-$k$ consistently has smaller sample complexity (up to 3 times smaller). We will put more emphasis on this last finding in our revision, which is currently only briefly mentioned in l.302-303 of the main paper.
>
>  * We are not sure of what you mean by ``considering the variance of the distributions in the definition of the Pareto front''. Right now, our goal (as that of prior work) is to identify the Pareto set of the set of means vectors, assuming that the marginal distributions of all objectives are sub-Gaussian with a known common bound on their sub-Gaussian parameter (l. 119-121). The setting and algorithms can be adapted to consider marginals that have different scales (i.e. different *known* bounds on their sub-Gaussian parameter), which is the case in our practical application, as explained in Appendix H.1. This amounts to identify the Pareto set of the vectors
> $\\{(\mu_i^{d}/\sigma^{d})_{d \in [D]}\\}_i$ where $\sigma^{d}$ is the sub-Gaussian parameter of criterion $d$.  But maybe what you had in mind is to consider unknown variances and possibly a risk-averse version of the PSI problem, e.g. in which the goal is to identify the Pareto set of the set  $\\{\boldsymbol{\mu}_i - \alpha \boldsymbol{\sigma}_i^2\\}_i$  for some parameter $\alpha > 0$, where the vector $\boldsymbol{\sigma}_i^{2}$ contains the variance of each objective $d$ (which could now depend on $i$ as well). Adapting our algorithm to this setting would require significant effort (e.g. building confidence intervals on unknown variances) that are beyond the scope of this paper.

---

> > ### Comment · Reviewer_J7Kd · 2023-08-18
> > **Thanks.**
> >
> > Thanks for the rebuttal. I keep my original score.

---

### Official Review · Reviewer_9Lr4 · 2023-07-10

**Soundness:** 3 good
**Presentation:** 3 good
**Contribution:** 2 fair
**Rating:** 6
**Confidence:** 3

**Summary:**

This paper studies a relaxed problem of Pareto set identification, in which the learner is required to identify a subset of the optimal arms. A single sampling strategy APE is proposed and then combined with various stopping rules to realize different relaxations of the original problem. In theory, this paper derives the sample complexity of these combinations. The proposed method is also validated empirically on a real-world scenario of Covid-19 vaccination.

**Strengths:**

1. Relaxed Pareto set identification is an important problem for multi-objective MAB. The proposed new framework may inspire new researches in this field, if properly justified.

2. The proposed sampling strategy sounds novel, and the analysis is technically sound.

3. The experiment on selecting vaccination strategies against Covid-19 is interesting.

**Weaknesses:**

1. One of the main contributions of this paper is to propose a new problem of $\epsilon$-PSI-$k$ and provide an analysis framework for this problem. However, I think the motivation of $\epsilon$-PSI-$k$ should be explained in more detail. I am not sure if we need this new formulation as $(\epsilon_1,\epsilon_2)$-PSI has already dealt with the sparsity issue.

2. Some detailed explanations of the intuition of the proposed sampling strategy (and why it outperforms previous strategies in principle) may help better understand the novelty in the algorithm design.

**Questions:**

See the weaknesses.

**Limitations:**

Limitations have been properly discussed.

---

> ### Author Rebuttal · Authors · 2023-08-08
>
>
> Thank you for your review. We address your two concerns about weaknesses below.
>
> * Our motivation for the $\epsilon_1$-PSI-$k$-relaxation comes from possible applications to early stage clinical trials, e.g. in vaccinology as in our real-world scenario. In this context, the constraint to identify at most $k$ interesting arms that will be investigated in further phases of clinical development (phase II, phase III) comes from material constraints: the cost of producing distinct treatments at a larger scale, a maximum number of patients that has to remain an order of magnitude larger than the number of arms, etc. (see also line 47-53 in our introduction). It is true that the parameter $\epsilon_2$ in the $(\epsilon_1,\epsilon_2)$-PSI relaxation also enforces some sparsity, but it provides no control on the size of the output, which is desirable in the above mentioned scenario. Thus, we view these two relaxations as two different ways of ensuring sparsity, and one of our main contribution is to propose a single exploration strategy (APE) that can tackle both objectives. Our strongest results are for $\epsilon_1$-PSI-$k$, for which we manage to quantify the reduction in sample complexity resulting from the limitation $k$ on the size of the output, both in theory and in practice.
>
>  * Regarding the novelty in algorithmic design, we emphasize that our sampling rule is the first fully sequential sampling rule ever proposed for (any of the studied relaxations of) Pareto front identification. It is fully sequential in the sense that in each round $t$, the most informative arm $a_t$ is selected in a data-dependent way. All prior algorithms are based on a ``racing'' approach, i.e. they use uniform sampling and an accept/reject mechanism. It is known in single objective bandits that fully sequential algorithms usually outperform their racing counterparts (see [9]) and we extend this observation to the multi-objective setting. Following your suggestion, we will provide a more detailed explanation of the intuition for our sampling rule, which is currently a bit shallow. It is built in the spirit of the adaptive lower/upper confidence bounds type algorithms in the single objective bandit setting, such as LUCB, LUCB++ or UGap. These three algorithms identify in each round two contentious arms: $b_t$: a current guess for the optimal arm (defined as the empirical best arm or smallest upper bound on its sub-optimality gap), $c_t$: a contender of this arm; the arm which is the most likely to outperform $b_t$ (in all three algorithms, it is the arm with the largest upper confidence bound in $[K]\backslash \\{b_t\\}$). Then, either both arms are pulled (LUCB, LUCB++) or the least explored among $b_t$ and $c_t$ is pulled. The originality of our sampling rule lies in how to appropriately define $b_t$ and $c_t$ for the multi-objective setting. The intuition for their definition is the following. Let $i$ be a fixed arm. Note that $M(i,j)>0$ for some $j$, if and only if there exists a component $d$ such that $\mu_i^d > \mu_j^d$ (recall that $M(i,j):= \max_d (\mu_i^d - \mu_j^d)$) i.e $i$ is not dominated by $j$. Moreover, the larger $M(i,j)$, the more $i$ is non-dominated by $j$ in the sense that there exists $d$ such that $\mu_i^d >> \mu_j^d$. Therefore, $i$ is strictly optimal if and only if  for all $j\neq i$, $M(i,j)>0$ i.e $\alpha_i:= \min_{j\neq i}M(i,j)>0$. And the larger $\alpha_i$, the more $i$ looks optimal in the sense that for each arm $j\neq i$, there exists a component $d_j$ for which $i$ is way better than $j$. As the $\alpha_i$'s are unknown, we define $b_t$ as the maximizer of an optimistic estimate of $\alpha_i$. We further restrict the maximization to arms for which we are not already convinced that they are optimal (by Lemma 1, the arms in $OPT^{\varepsilon_1}(t)$ are (nearly) Pareto optimal on the event $\mathcal{E}$). Then, we note that for a fixed arm $i$, $M(i,j) < 0$ if and only if $i$ is strictly dominated by $j$. And the smaller $M(i,j)$, the more $j$ is close to dominate $i$ (for any component $d$, $\mu_i^d - \mu_j^d$ is small). Thus, for a fixed arm $i$, $\text{argmin}_{j\neq i} M(i,j)$ can be seen as the arm which is the closest to dominate $i$ (or which dominates it by the largest margin).
>  By minimizing a lower confidence bound on the unknown quantity $M(b_t,j)$, our contender $c_t$ can be interpreted as the arm which is the most likely to be (close to) dominating $b_t$. Gathering information on both $b_t$ and $c_t$ can be useful to check whether $b_t$ can indeed be optimal.

---

### Decision · Program_Chairs · 2023-09-21

**Decision:**

Accept (poster)

**Comment:**

Bandits problems with multiple objectives is a timely topic. The paper proposes a new theoretical framework to pose concrete questions in the form of several relaxations of PSI (Pareto Set Identification), the mutli-objective analog of Best-Arm-Identification (BAI). The theoretical results seems strong, novel and interesting and are complemented by an empirical study.